# A Variable Attention Nested UNet++ Network-Based NDT X-ray Image Defect Segmentation Method

**Jiayin Liu**  **and Jae Ho Kim ***

Image and A.I. Laboratory, Department of Electronics Engineering, Pusan National University, Busan 46241, Korea; liujiayinpnu@gmail.com
* Correspondence: jhkim@pusan.ac.kr; Tel.: +82-10-4042-2450

**Abstract:** In this paper, we describe a new method for non-destructive testing (NDT) X-ray image defect segmentation by introducing a variable attention nested UNet++ network. To further enhance the performance of the faint defect extraction and its clear visibility, a pre-processing method based on pyramid model is also added to the proposed method to effectively perform high dynamic range compression and defect enhancement on the 16-bit raw image. To illustrate its effectiveness and efficiency, we applied the proposed algorithm to the X-ray image defect segmentation problem and carried out extensive experiments. The results support that the proposed method outperforms the existing representative techniques in extracting defect for real X-ray images collected directly from industrial lines, which achieves the better performance with 89.24% IoU, and 94.31% Dice.

**Keywords:** defect segmentation; X-ray image; variable attention; UNet++

## 1. Introduction

A variety of non-destructive testing (NDT) methods are used in industrial production, to detect the internal defects of objects, among which NDT methods based on X-ray imaging are most widely used, mainly because of their intuitive imaging and high spatial resolution of imaging, especially for the detection of small defects in complex structural products [1–3]. At present, the use of industrial film is still very large. Its disadvantages include slow imaging and being not environmentally friendly, and the process of preliminary preparation, exposure, film processing, interpretation, etc. is very time-consuming. For complex workpieces, it takes several hours of inspection time to complete the whole process, and most importantly, it is difficult to digitize, which brings difficulties to the storage and retrieval of film. With the development of industrialization, a large number of products need full inspection. The previous use of film imaging for random inspection can no longer meet the current pace of inspection, and so the industry of NDT X-ray imaging has an urgent need for digitalization and automation.

Early digital X-ray imaging technology uses an image intensifier (I.I.) as the imaging device, which can output a digital image of up to 12-bits, using the intensifier to output an analog signal and then a digital camera to enhance and digitize the analog signal. Spatial resolution and contrast sensitivity are the most important imaging indicators in X-ray imaging. However, these two indicators of the image intensifier imaging system are not up to the level of film imaging; coupled with the lack of relevant testing standards to support, its application in the field of industrial inspection is limited. In actual industrial applications, large-scale use of image intensifiers for defect detection occurs only in two fields: automotive wheel castings [4,5] and some electronic devices, and small-scale use of the field of welding seams [6] and cylinder inspection [7]. Other areas with strict requirements for detection choose to use film imaging.

The latest X-ray imaging technology uses digital flat panel detectors as imaging devices, which can output 16-bit digital X-ray images with contrast sensitivity indexes

that can exceed film imaging. In recent years, the price of flat-panel detector hardware has been reduced to a level generally accepted by industry, and the development of deep learning-based correlation segmentation and identification algorithms has made automated X-ray imaging-based detection possible. However, the diversity of inspected products and their respective complex structures, as well as small and ambiguous defect structures, make automatic defect detection on 16-bit image data extremely complex and challenging.

Defect detection can be roughly divided into three steps: (a) locating and segmenting defects on the X-ray digital image of the inspected product, (b) identifying and quantitatively analyzing the segmented defects, and (c) judging whether the product is qualified according to the relevant analysis data combined with the inspection process requirements. Step (a) is the most difficult one, and it is very important to accurately and completely segment the defects from the image containing the complex structure of the inspected product for later quantitative analysis.

In general, methods for automatic defect segmentation can be divided into two categories: unsupervised and supervised. A variety of unsupervised techniques are used to extract target defects from complex backgrounds [8,9], including matched filtering, morphological processing, defect tracking, etc. One advantage of unsupervised segmentation methods is that no sample annotation is required; however, the practical performance of these methods is not good, especially for small-sized defects with more blurred edges [10]. However, most of the methods in unsupervised learning are based on traditional image processing algorithms, which are not ideal for detection when encountering various complex and variable practical applications, because it is difficult to extract effective features and summarize prediction rules. For example, in the paper [3], the authors used traditional image processing algorithms, mainly median filter and morphological processes for segmentation of defects. The experimental part of the data is relatively small and does not work well on images with slightly complex backgrounds. In [11,12] the authors used the SDD segmentation algorithm to segment ventricles in MRI images and cells or nanoparticles in microscopic images, which have the best performance among the listed single-threshold segmentation methods. However, due to the complicated structure of the workpieces, the grayscale distribution of image background is not uniform. It can be inferred that methods in [11,12] are not very applicable for the industrial X-ray defect segmentation tasks. For supervised methods, the sample image first needs to be manually labeled to mark the defects in it [13], and then the features are separated into background and defects using a trainable classifier. In most cases, the supervised methods perform better than the unsupervised based methods [14,15]. A recent review of deep learning for general object detection can be found in [16], where the authors provide a comprehensive survey of the recent achievements about deep learning for generic object detection.

In recent years, with the rise of deep learning research, several researchers have introduced supervised methods based on deep learning to the task of defect segmentation in X-ray images [17,18]. A very basic fully convolutional network (FCN) for weld defect identification is presented in [19], in which the author illustrates some inspection results on the GDXray database, but comparisons to other deep learning methods and more in-depth tests are lacking. The U-Net algorithm is built on an FCN, consisting of an encoder and a decoder, and the shape of the network resembles a "U" shape, hence the name "U-Net". U-Net is very different from other common segmentation networks in that U-Net uses a completely different feature fusion method: splicing, where U-Net splices features together in the channel dimension to form thicker features [20–22].

Subsequently, the better-performing UNet++ was developed based on U-Net [23]. UNet++ improves segmentation accuracy through a series of nested, dense jump paths that meet the high accuracy requirements for defect detection [24]. The redesigned jump paths make it easier to optimize feature mapping with semantically similar features. Dense jump connections improve segmentation accuracy and improve gradient flow. Deep supervision allows model complexity tuning to balance speed and performance optimization.

UNet++ has been widely used in biological image segmentation, such as retinal vascular segmentation, liver CT image segmentation [25], lung CT image segmentation [26], ultrasound medical image segmentation [27], COVID-19 infection localization [28], heart CT image segmentation [29], etc., and has achieved good results. There are also applications in other scientific and industrial fields, such as the detection of impact craters on the lunar surface [30], road detection [31], etc.

In recent years, there are also some other excellent deep learning network structures for detecting objects in a variety of application scenarios. Vgg16 is a CNN network that simply superimposes convolutional or fully connected layers with weights to 16 layers. As the size of the input image is limited to $224 \times 224 \times 3$, it is difficult to detect smaller defects at the pixel level, and only locates out with boxes for weld defects, with no quantitative output [32]. A spatial attention bilinear convolutional neural network (SA-BCNN) was introduced and tested against other CNN based methods [33]. Faster R-CNN is recognized to have better performance and much research has been done by many scholars. Based on a basic Faster R-CNN system, Feature Pyramid Network (FPN) shows significant improvement as a generic feature extractor in several applications [34]. It achieved state-of-the-art single-model results on the COCO detection benchmark without bells and whistles, surpassing all existing single-model entries including those from the COCO 2016 challenge winners at the time. A method based on Feature Pyramid Network (FPN) and its subsequent improvements was used to detect defects in radiographic images of casting aluminum parts [35,36]. The experimental results outperformed Faster R-CNN; it was an instance segmentation method that could not segment the defects at pixel level and thus could not produce quantitative defect detection results. A very similar pyramid approach is used in networks for deep learning [37], wherein the spatial pyramid pooling is used to remove the uniform limitation on the size of the input image; the pyramid method is a common and useful tool for adapting to different resolutions and scales. Mask R-CNN is introduced through extending Faster R-CNN by adding a branch for predicting an object mask in parallel with the existing branch for bounding box recognition so that it can efficiently detects objects in an image while simultaneously generating a high-quality segmentation mask for each instance, which means it is more suitable for quantitative defect detection [38].

There is no available large-scale defects database of X-ray images, and many scholars use Generative Adversarial Networks (GAN) to generate more defect data for learning in networks such as CNN. A CNN-based method for X-ray prohibited item recognition has been proposed [39], and additionally, generative adversarial networks (GANs) are used for data augmentation. In another CNN-based casting defect detection work [40], the author builds the dataset by using synthetic defects, which are simulated using 3D ellipsoidal models and Generative Adversarial Networks (GAN). This is done not only for X-ray images: in [41] the authors also use GAN to generate more visible defect images to improve defect detection.

In this paper, we propose a novel variable attention-based nested segmentation network that improves the lower segmentation accuracy of the standard UNet++ network using fixed perceptual field convolution. It can automatically adjust the perceptual field of the network by the attention mechanism to more effectively utilize the spatial information extracted at different scales and introduces an attention mechanism between the nested convolutional blocks so that features extracted at different levels can be selected for merging relevant to the segmentation task, as a way to improve the defect segmentation effect of the whole network on sample images with complex background structures. In addition, a new image pre-processing algorithm based on the pyramid model is proposed in this paper, which effectively performs high dynamic range compression and defect enhancement on the original 16-bit images. The grayscale distribution of the processed image is more balanced, and the faint defects that are not easily detected by human eyes are clearly visible, which is convenient for manual labeling.

We further describe the comparison between the proposed method and other related typical methods involved, and to see the advantages and disadvantages of each method more clearly, we summarize them in a table as follows in Table 1.

**Table 1.** The comparison between the proposed method and other industrial X-ray defect detection methods.

| Types | Methods and Brief Description | Advantages | Disadvantages |
|---|---|---|---|
| Unsupervised | Median and morphological filters, Ref [12] | Very easy to implement. | Does not work well on images with slightly complex backgrounds. |
| | Defect detection based on traditional algorithms, Ref [4] | Better performance than simple morphological and threshold. | Poor results on complex background images. |
| Supervised | LBP descriptor with an SVM-linear classifier, Ref [16] | Simple features and framework structure. | Not a pixel-level defect segmentation. |
| | U-Net, Ref [21] | Thicker features, defect fusion at different scales. | The evaluation score of 80% indicates that the system needs modification for better performance. |
| | Proposed method | Developed HDR pre-processing to enhance more details, large quantity and huge variety of images for training | A little bit time-consuming; it is still acceptable for practical applications. |

The experimental results show that after combining the image pre-processing algorithm, the variable attention-based nested UNet++ network proposed in this paper has better detection effect and higher accuracy for X-ray image defects than the other selected network. It was observed that the proposed segmentation method exhibits the top segmentation performance, which holds the leading position with 89.24% IoU, and 94.31% Dice.

## 2. Methods Section: Algorithm

Using the Pytorch deep learning framework, a variable attention-based nested segmentation network with selective kernel convolution was built. The structure of the proposed network in this paper is based on the U-Net++ network with a nested architecture. For the convolutional module in the network, an improved SK module was used instead of the traditional ordinary convolutional module, and an attention mechanism is introduced between the nested convolutional blocks of the network so that features extracted at different levels can be selectively merged to improve the efficiency of propagating semantic information through jump connections. The proposed network suppresses background regions that are irrelevant to the segmentation task, while having the ability to increase the weight of the target region, which in turn improves the accurate segmentation of defects.

The flow chart of the entire defect detection method is shown in Figure 1, the main important process is divided into: input image, pre-processing, automatic defect detection, and output detection result. The detailed description of each module will be detailed in the subsequent part of this chapter.

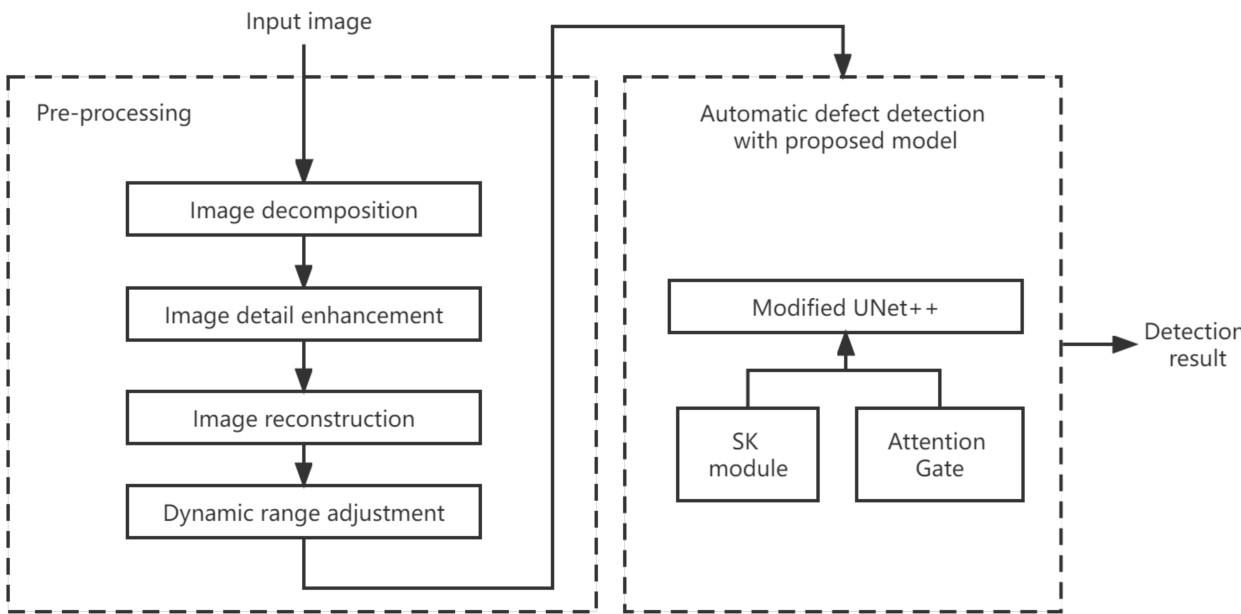

**Figure 1.** The flow chart of the entire defect detection method.

*2.1. Image Pre-Processing*

The original image acquired by the X-ray flat-panel detector is a 16-bit grayscale image with a large dynamic range, and the difference between the grayscale values of the defects themselves and those of the background is small, and the grayscale distribution of the background is not uniform due to the inherent X-ray beam-hardening imaging characteristics. We propose a new image pre-processing algorithm based on the pyramid model to effectively perform high dynamic range compression and defect enhancement on the 16-bit raw image. The grayscale distribution of the processed image is more balanced, and the faint defects that are not easily detected by human eyes are clearly visible, which is convenient for manual labeling.

The core idea of our proposed pre-processing algorithm is to decompose the image into pixels representing individual details of the image, then do enhancement on these pixels, and finally perform inverse reconstruction. We choose the Laplace pyramid function for image decomposition, which meets the following two basic conditions: (1) it must include all levels to represent the structure of any size, and (2) it must be continuous without interruption. The effect of pre-processing is shown in Figure 2. On the left is the original image, and on the right is the pre-processed image, where the faint porosity defects in the weld are clearly visible after processing.

2.1.1. Image Decomposition and Reconstruction

The basic idea of Laplace pyramid function decomposition is that first, the original image is low-pass filtered to reduce the closeness of the pixel-to-pixel connection, interval sampling compresses the image data, which means the image sample density is reduced, then interpolation is performed, and finally the resulting image is subtracted from the original image as the first layer in the Laplace pyramid. Repeating the above operations based on this layer of images expands into a pyramid-shaped multi-scale data structure.

Laplace pyramid is built on Gaussian pyramid and consists of a series of L0, L1, L2, L3, L4, etc. As shown in Figure 3, each L is the set of differences between two adjacent Gaussian pyramids, i.e.,

$$L\ (t) = G\ (t) - \text{expand}\ (G\ (t + 1)), \tag{1}$$

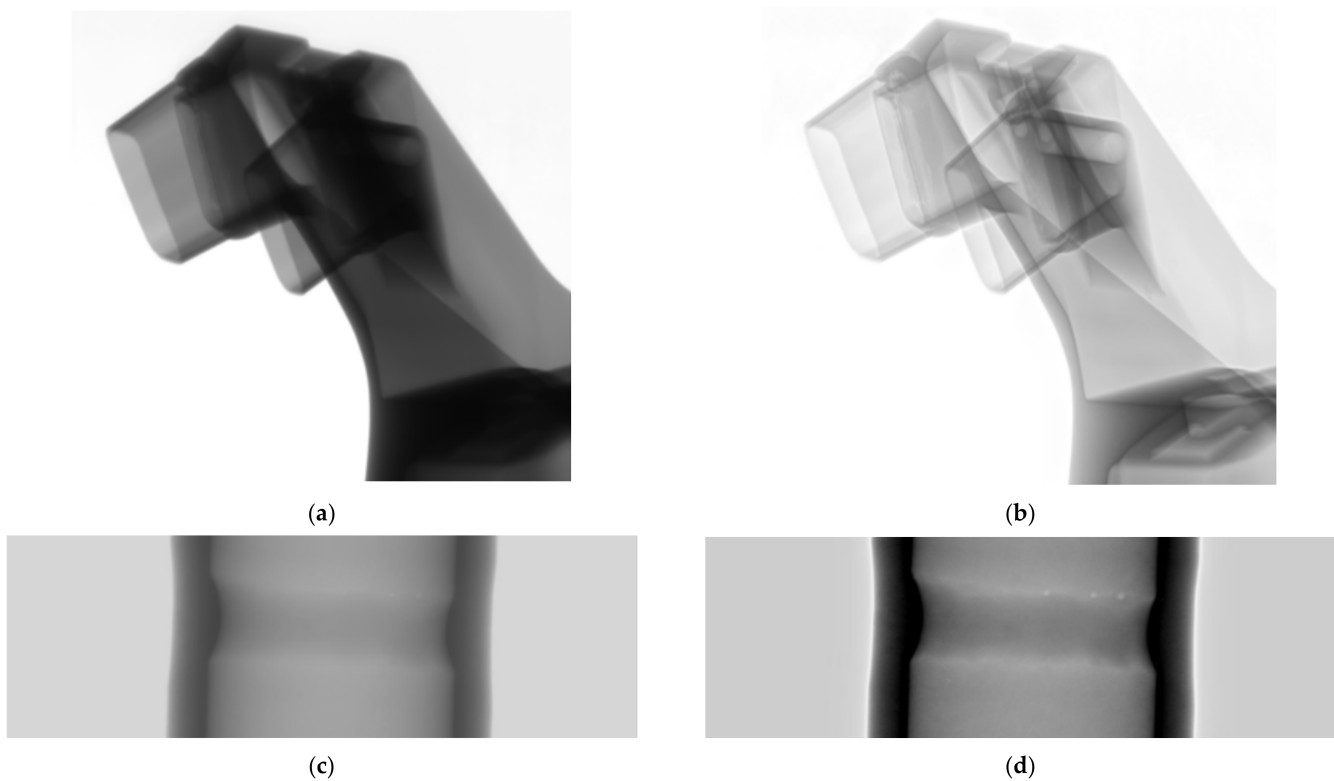

**(a)**　　　　　　　　　　　　　　　　**(b)**

**(c)**　　　　　　　　　　　　　　　　**(d)**

**Figure 2.** (**a**) Original casting image, (**b**) pre-processed image of (**a**), (**c**) original weld image, (**d**) pre-processed image of (**c**).

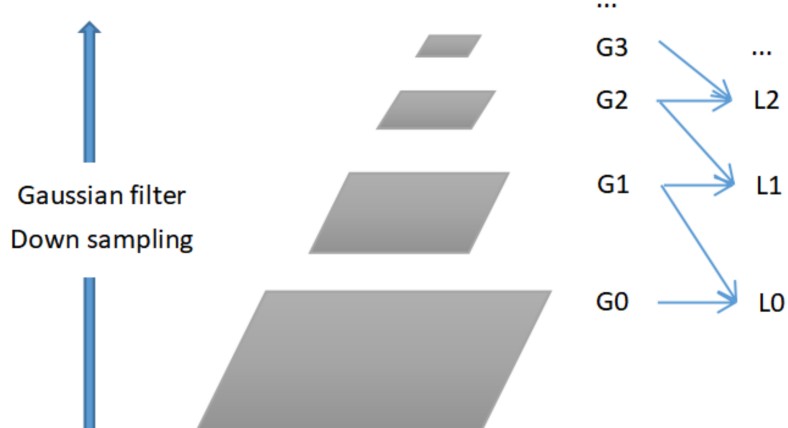

**Figure 3.** Laplace Pyramid Structure. G0, G1, G2, G3 are the Gaussian pyramid images. In a Gaussian pyramid, subsequent images are weighted down using a Gaussian average and scaled down. L0, L1, L2 are the Laplace pyramid images which save the difference image of the Gaussian images between each levels.

In the image decomposition process, each image is reduced by half (i.e., reduced to one-half of the original sample density) from the previous one, at which point the whole process presents a pyramid data structure. During image reconstruction, the Gaussian pyramid needs to be scaled up and added to the Laplace pyramid of a lower level, i.e.,

$$G(t) = L(t) + \text{expand}(G(t+1)), \tag{2}$$

2.1.2. Image Detail Enhancement

The main purpose is to enhance the details of the image by adjusting the spatial frequency characteristics of the image to highlight the subtle defective features in the image, specifically to achieve

$$Y = X + a \times B(X) \times (X - X1), \qquad (3)$$

where Y, X, and X1 represent the pixel values of the resultant image, the original image, and the image after low-pass filtering, $(X - X1)$ represents the high-frequency part of the image, and the coefficient a determines the degree of enhancement of the high-frequency part.

First, the original image is low-pass filtered, that is, smoothed to obtain a smooth image, and then the difference between the original image and the smoothed image to obtain the difference image. The difference image represents the high-frequency information part of the image, usually the edge and detail information part of the image, according to the different data density parts of the image for the corresponding degree of image enhancement, where the coefficient a, function B (X) is given in advance. Here, the coefficient a and function B (X) are given in advance, and will vary according to the image type, image effect requirements, and image data density distribution. The image data density distribution in the algorithm is particularly important, because in the later algorithm Laplace pyramid decomposition, hierarchical enhancement coefficients are related to the density of the image data.

2.1.3. Reduced Dynamic Range

The dynamic range of the image is reduced in the low-frequency part, and the density compensation of the region of interest is implemented:

$$Y = X + a \times (A - X1) \qquad (4)$$

where X denotes the original image, X1 denotes the smoothed and filtered image, a is the correction factor where a < 1 (used to control the degree of dynamic compression), and A is a constant.

Firstly, the original image is smoothed by low-pass filtering to obtain a smoothed image, then the difference is made with the given A to obtain a difference image, and then multiplied with the given coefficient a. The meaning is to perform dynamic compression of image data density, i.e., the low frequency part of the image is removed and the high frequency part is retained, because the important detail information of the image often exists in the high frequency. The final image obtained is compared with the original image. The final image is obtained by summing with the original image.

*2.2. Defect Segmentation Network*

The Pytorch deep learning framework is used to build this network, and the structure of the proposed convolutional neural network in this paper is shown in Figure 4, which is based on the UNet++ network and uses a nested architecture to integrate U-Net of different depths.

The network designed in this paper nests four layers of U-Net as the basic network framework, where encoders and decoders are symmetrically distributed on both sides of the network. All layers of U-Net share one feature extractor so that we only need to train one encoder. A modified SK block is used to replace the traditional convolutional block in the network, and a convolution with a perceptual field of 5 is generated by using two $3 \times 3$ convolutions in series in the SK block, which both improves the depth of the network and reduces the computation and number of parameters, as shown in Figure 5. By using the SK block, the perceptual field can be automatically adjusted to make more efficient use of the feature information extracted at different scales. The encoder has a total of five layers, each of which consists of two modified SK blocks + Relu. Each layer undergoes a maximum pooling of size $2 \times 2$ with a step size of 2 after feature extraction. Each subsequent layer of the structure is down-sampled in the same order.

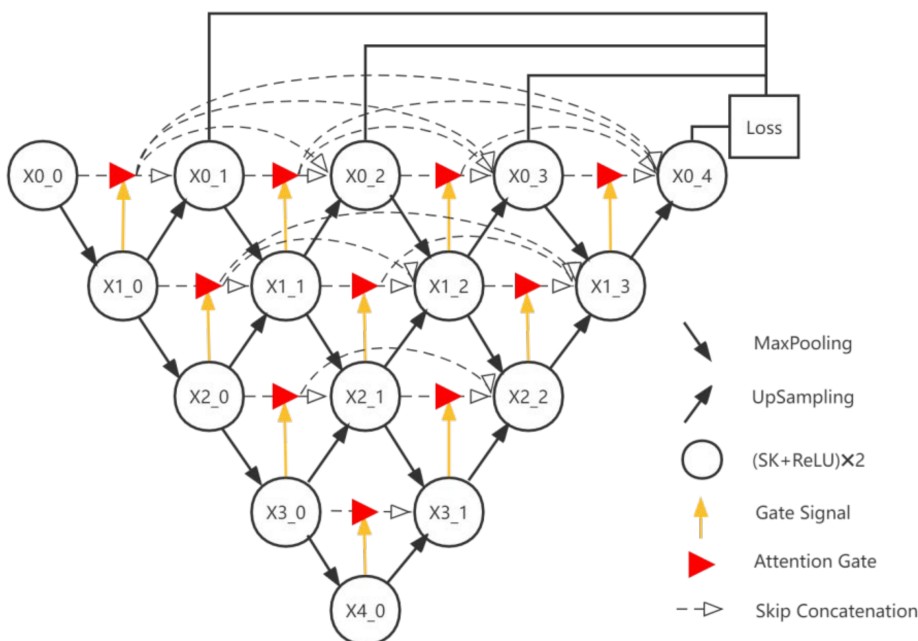

**Figure 4.** Schematic diagram of the structure of the proposed defect segmentation network. SK block is adopted instead of convolutional block with an attention gate.

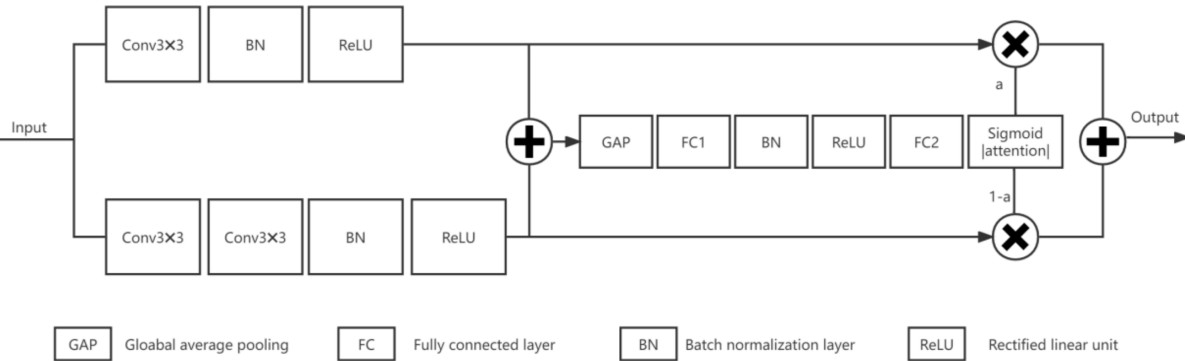

**Figure 5.** Schematic diagram of the structure of the improved SK module. The SK block consists of two branches. The first one utilizes one conventional $3 \times 3$ convolutions, while the second one uses two $3 \times 3$ convolutions to generate a perceptual field of 5.

To focus on features related to the target or goals, we add a simple but effective attention gate to the nested architecture, as shown in Figure 6. This attention gate has two inputs: an up-sampled feature Fg in the decoder and a feature Fx of equal depth in the encoder. The selected signal Fg in the attention gate selects the more useful features from the encoded feature Fx and sends them to the upper decoder.

The contextual information extracted by the encoder is propagated to the decoder of the corresponding layer through a dense jump connection, thus allowing the extraction of more efficient layered features. In the case of dense jump connection, the input of each convolutional block in the decoder consists of two equal-scale feature maps: (1) the intermediate feature map from the output of the previous potential gate along the same depth of the jump connection; and (2) the final feature map from the output of a deeper deconvolution block operation. After receiving and concatenating all the to-be-feature maps, the decoder recovers the image in a bottom-up manner.

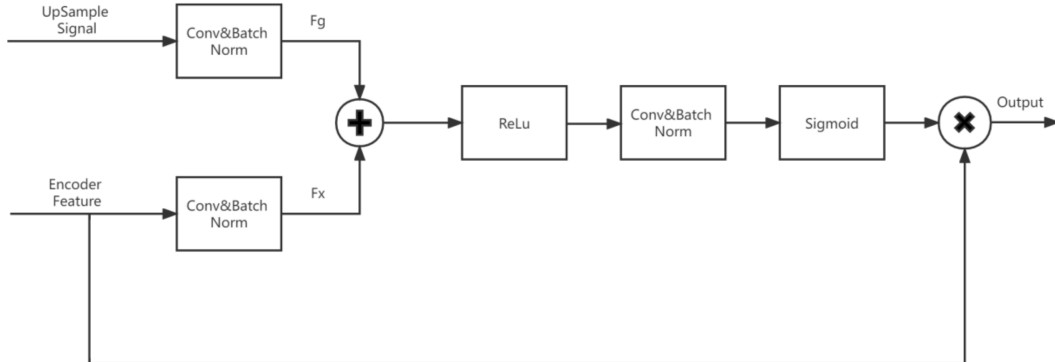

**Figure 6.** Schematic diagram of the structure of the attention gate. Encoder features are scaled with attention coefficients calculated in the attention gate.

We define the feature mapping to represent the output of the convolutional block, where i represents the depth of the feature in the network and j represents the sequence of convolutional blocks in layer i connected along the jump, as follows.

$$X_{i,j} = \begin{cases} \phi[X_{i-1,j}] & j = 0 \\ \phi\left[\sum_{k=0}^{j-1} Ag(X_{i,k}), UP(X_{i+1,j-1})\right] & j > 0 \end{cases}, \tag{5}$$

where $\phi[]$ denotes the concatenated merging of convolutional blocks. $UP[]$ and $Ag[]$ denote up-sampling and attention gate selection, respectively.

Deep supervision is introduced in the network structure by attaching a $1 \times 1$ convolution with a C kernel and a sigmoid activation function to the outputs of nodes X0_1, X0_2, X0_3 and X0_4, where C is the number of classes in which a given dataset is available. A hybrid segmentation loss is then defined for each semantic scale, including pixel-level cross-entropy and Dice coefficient loss. The hybrid loss can take advantage of the two loss functions: smooth gradients and class imbalance handling. It is defined as follows.

$$L(Y, P) = -\frac{1}{N} \sum_{c=1}^{C} \sum_{n=1}^{N} \left( y_{n,c} \log p_{n,c} + \frac{2y_{n,c} p_{n,c}}{y_{n,c}^2 + p_{n,c}^2} \right), \tag{6}$$

of which $y_{n,c} \in Y$ and $p_{n,c} \in P$ denote the target labels and predicted probabilities for class $c$ and $n$ pixels in a batch, and $N$ denotes the number of pixels in a batch.

Because the output of each sub-network is already the segmentation result of the image during deep supervision, we can cut out those redundant parts if the output result of the smaller sub-networks is already good enough.

The main advantages of the defect segmentation method proposed in this paper include the following:

1. The traditional convolutional blocks are replaced by SK blocks, and the convolutional blocks with a perceptual field of 5 in the SK blocks are replaced by two 3x3 convolutions in series, which not only improve the depth of the network but also reduce the computation and the number of parameters. By using the SK block, the perceptual field can be automatically adjusted to utilize the feature information extracted at different scales more effectively.
2. Adding attention gates between nested convolutional blocks enables increase of the weight of the target region while suppressing background regions that are not relevant to the segmentation task.
3. It enables model pruning during testing by introducing deep supervision, which can reduce a large number of model parameters and thus speed up the model segmentation.

### 3. Experimental Section

In this section, both quantitative and qualitative results are reported with an extensive set of comparative evaluations for defect segmentation.

### 3.1. Experimental Equipment and Database

Unlike the vast database of publicly available medical X-ray images, currently, the only publicly accessible X-ray imaging database for industrial NDT is GDXray, which was published by D. Mery in 2015 [42]. The database includes five groups of X-ray images: castings, welds, baggage, natural objects, and settings. However, the image number of GDXray's database is too small, with only 2727 images from different angles on 67 samples of castings, and only 88 images from different positions on three samples of welds. In addition, most of the images were taken with the Image Intensifier (I.I.) and saved in 8-bit BMP or JPG format. Thus, the image quality and spatial resolution were not high enough, and could not be adapted to the contemporary needs of industrial inspection. Figure 7 shows almost 40 images generated at different shooting angles from one single aluminum casting wheel in GDXray's database. Therefore, we needed to build our own database.

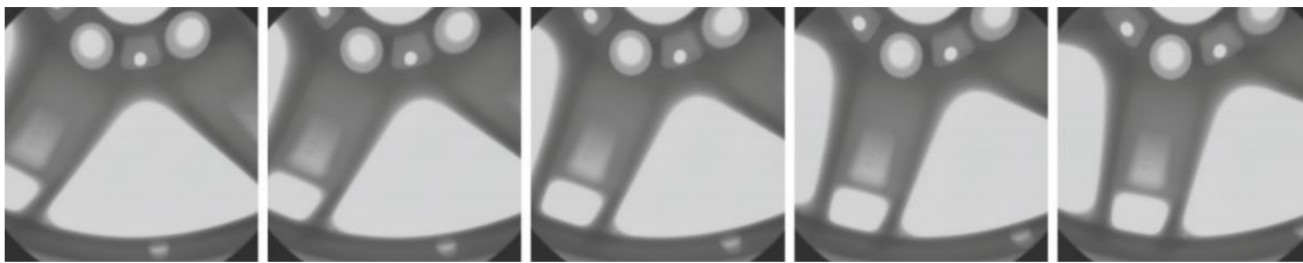

**Figure 7.** Selected images from GDXray's database [42].

We have cooperated with Deepsea Precision Co. Ltd. (Shenzhen, China) [43], a manufacturer specializing in X-ray inspection equipment, to collect a large amount of high-quality, high-resolution X-ray image data from actual production lines and laboratories over a period of three years. As shown in Figure 8, this is industrial X-ray inspection equipment from Deepsea Precision Co. Ltd., and our database was collected from this equipment and more than a dozen other similar machines. The basic configuration of the core imaging chain components from this typical X-ray inspection equipment of Figure 8a is shown in Table 2.

**Table 2.** The basic configuration of the core imaging chain components from the X-ray inspection equipment of Figure 8a.

| Device Name | Brand/Model | Basic Configuration |
|---|---|---|
| X-ray emission device (macro-focus) | Gulmay/CF500 | 500 kV, focus size 0.4/1.0 mm |
| X-ray emission device (micro-focus) | WorX/XWT-225-CT | 225 kV, focal size 5 μm |
| X-ray receiver device (flat panel detector) | Deepsea/DS4343HR | 430 mm $*$ 430 mm, pixel size 139 μm |
| Workstation Software | Deepsea/DeepVISION | GPU-based architecture |

The database we created is named DSXImage, and its architecture is shown in Table 3. It is divided into two main types, castings and welds, and the image format is fully compliant with the industrial NDT standard ASTM E2339-15 [44] and is saved in 16-bit DICONDE format. The breakdown within each main category is as follows.

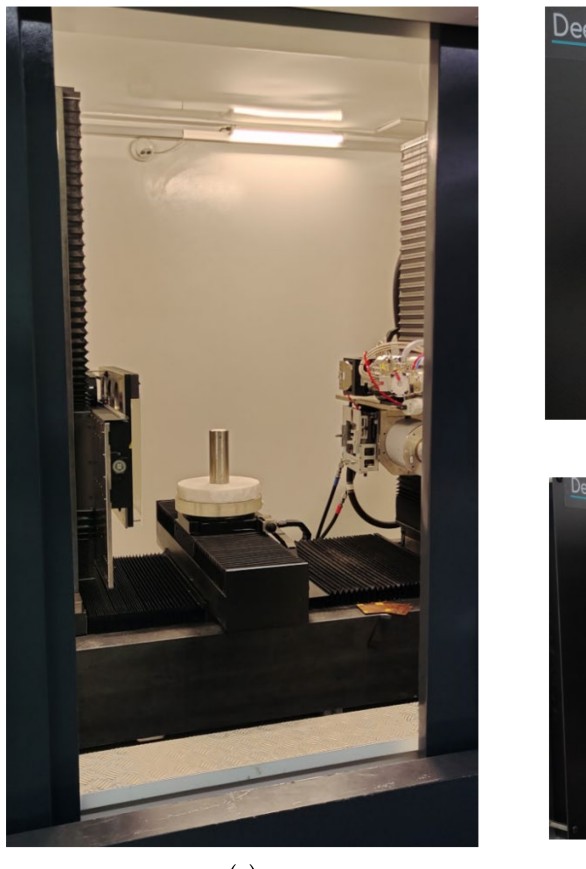

(**a**)

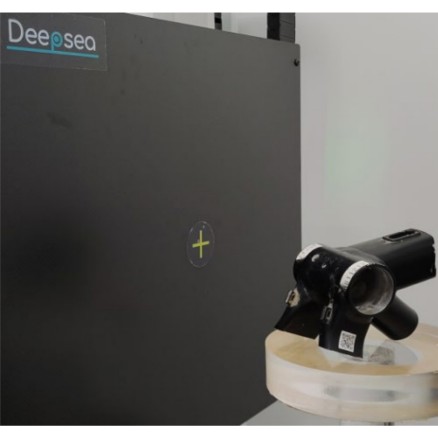

(**b**)

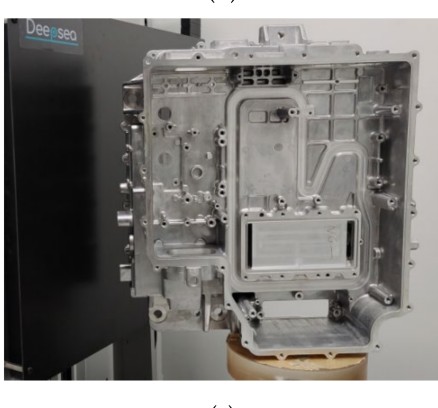

(**c**)

**Figure 8.** Some typical X-ray inspection equipment from Deepsea. (**a**) A 500 kV X-ray inspection machine, steel weld sample; (**b**) a 225 kV X-ray inspection machine, aluminum casting sample from a bicycle; (**c**) a 225 kV X-ray inspection machine, aluminum casting sample from automobile.

**Table 3.** DSXImage database architecture.

| Types | Sample Name | Amount |
|---|---|---|
| Castings | engine cylinder heads | 10,500 |
| | steering knuckles | 8602 |
| | shock absorbers | 15,610 |
| | valves | 9500 |
| | power supply housings | 6500 |
| | 3C products and others | 18,805 |
| Welds | rocket engine ducts | 12,655 |
| | aircraft seat bases | 5431 |
| | gas pipe welds | 25,820 |
| | steel welds from 3C and others | 12,907 |

1.  Most of the castings are aluminum alloy parts for automobiles, such as engine cylinder heads, steering knuckles, shock absorbers, valves, power supply housings, etc. A small portion are magnesium alloy parts for 3C products, such as Bluetooth headset metal frames, laptop bezels, etc.
2.  Welds are stainless steel/titanium alloy, collected from rocket engine ducts, aircraft seat bases, gas pipe welds, etc. A small portion are steel welds from 3C.

The DSXImage database is still in continuous improvement; however, the existing size of the database is sufficient for training and evaluation of deep learning networks. The database may be released to the public at the right time.

The project team worked with Deepsea to generate the DSXImage training dataset using the open-source labeling tool Labelme., which is a graphical image annotation tool inspired by MIT. The user can use it to easily implement image annotation work for vision tasks such as classification, detection, and segmentation, etc. The annotation result of one sample consists of three images and two label files. The architecture is summarized in Table 4. The image annotation work is a time-consuming and labor-intensive task that took nearly a year to label the entire database. The specific annotation process and output are summarized roughly as follows.

**Table 4.** The architecture of one annotation result, basic configuration of the core imaging chain components from the X-ray inspection equipment of Figure 7a.

| The Original Image | The Label Image | The Label Visualization Image |
| --- | --- | --- |

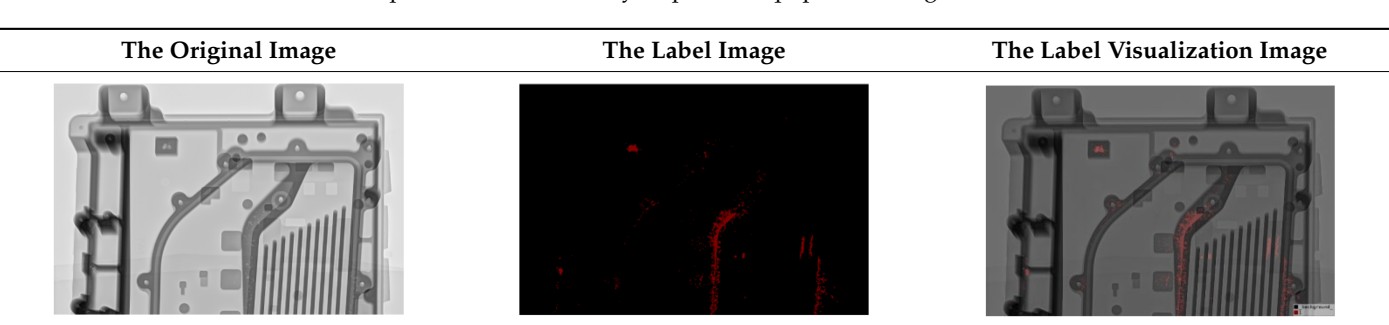

1.  Open the image, perform manual annotation, outline each defect and save it as a json label file.
2.  Open a json file and convert the json into a mask label image.
3.  After the conversion is completed, a label folder is generated, including the original image img.png, label image label.png, label visualization image label_viz.png, txt file of label name, and yaml format label name file.

The design principle of the database is to conform to the current mainstream image specifications and try to take into account the previous image specifications so that not only can previous algorithms use the database, but future algorithms can also be adapted by simple expansion and upgrade of the current database.

1.  Image bits. Most modern flat panel detectors output 16-bit images, older detectors are 12 or 14 bit, and earlier image intensifiers are 8 or 10 bit. The database uses the original 16-bit image with the addition of a pre-processed 16-bit image and an 8-bit image.
2.  Image resolution. Depending on the specifications of the flat panel detector, there is no uniform specification for the resolution of the images in the database. The pixel sizes of most images are 3072 × 3072, 1536 × 1536, 2000 × 2000, which users can choose according to their needs and can even be freely cropped if necessary.
3.  Image format. The 10-bit, 12-bit, 14-bit, and 16-bit images in the database use the industry standard image format, i.e., DICONDE format, and also keep a copy in TIFF format. 8-bit images are saved in PNG format.

*3.2. Experimental Setup*

The defect segmentation task was conducted over the DSXImage dataset comprising 126,330 sample X-ray images with corresponding image annotation. In all our experiments, we assumed an 80/20 split for train and test purposes respectively. Besides, 20% of training data was used as a validation set for model selection and to avoid overfitting. PyTorch library with Python 3.7 was used to train and evaluate the variable attention nested UNet++ network, running on a PC with Intel® Core™ i9-9900KF CPU at 3.6 GHz (Santa Clara, CA, USA), with 64 GB RAM, and with a 12 GB NVIDIA GeForce GTX 2060 GPU card (Santa Clara, CA, USA) [45,46].

Some of the original images in the DSXImage datasets are shown in Figure 9.

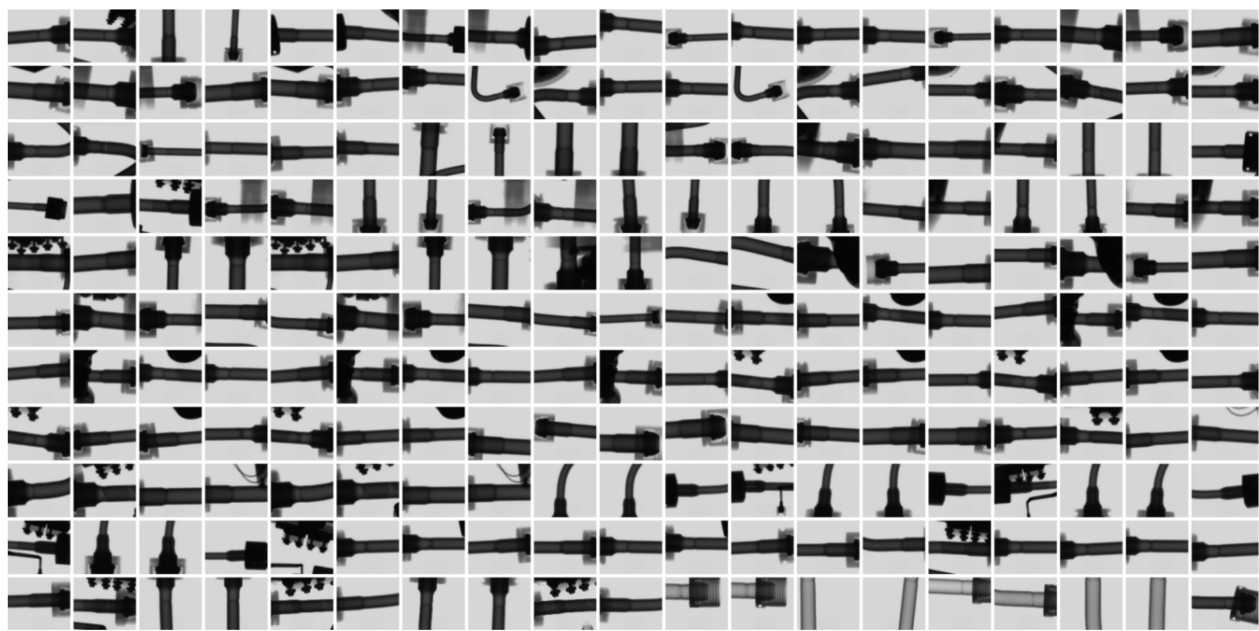

**Figure 9.** Some of the original images in the DSXImage dataset.

### 3.3. Segmentation Evaluation Metrics

The performance of the defect segmentation networks was evaluated using two evaluation metrics, namely, Dice (Dice Similarity Coefficient) and IoU (Intersection over Union) [28]. The corresponding equations are as follows.

$$Dice = 2 * TP/(2 * TP + FP + FN), \tag{7}$$

$$IoU = TP/(TP + FP + FN), \tag{8}$$

Here, TP, TN, FP, FN represent the true positive, true negative, false positive, and false negative, respectively. Both IoU and Dice are statistical measures of spatial overlap between the binary ground-truth and the predicted segmentation masks, where the main difference is that Dice considers double weight for TP pixels (true defect predictions) compared to IoU.

### 3.4. Segmentation Results

The performance of the all the selected segmentation methods over the test set is tabulated in Table 5. We have selected some classical and recent deep-learning-based segmentation algorithms with better performance. For all models, it was observed that the proposed segmentation method exhibits the top segmentation performance, holding the leading position with 89.24% IoU, and 94.31% Dice. Some examples of segmentation results obtained by different methods listed in Table 5 are shown in Figures 10 and 11. Figure 12 shows more examples of the segmentation results of the proposed method. Furthermore, the proposed method has been customized for different applications as illustrated in Figure 13.

**Table 5.** The performance of the all the selected segmentation method over the test set—DSXImage database.

| Models | IoU | Dice | #Parameters | Time per Epoch |
|---|---|---|---|---|
| Median and morphological filters, Ref [3] | 0.3012 | 0.4630 | – | – |
| Defect detection based on traditional algorithms, Ref [4] | 0.4155 | 0.5871 | – | – |
| Mask R-CNN | 0.7360 | 0.8479 | 25.3 M | 38 s |
| U-Net | 0.7251 | 0.8406 | 32.5 M | 16 s |
| Standard UNet++ | 0.7629 | 0.8656 | 35.6 M | 39 s |
| Proposed Method | 0.8924 | 0.9431 | 36.8 M | 42 s |

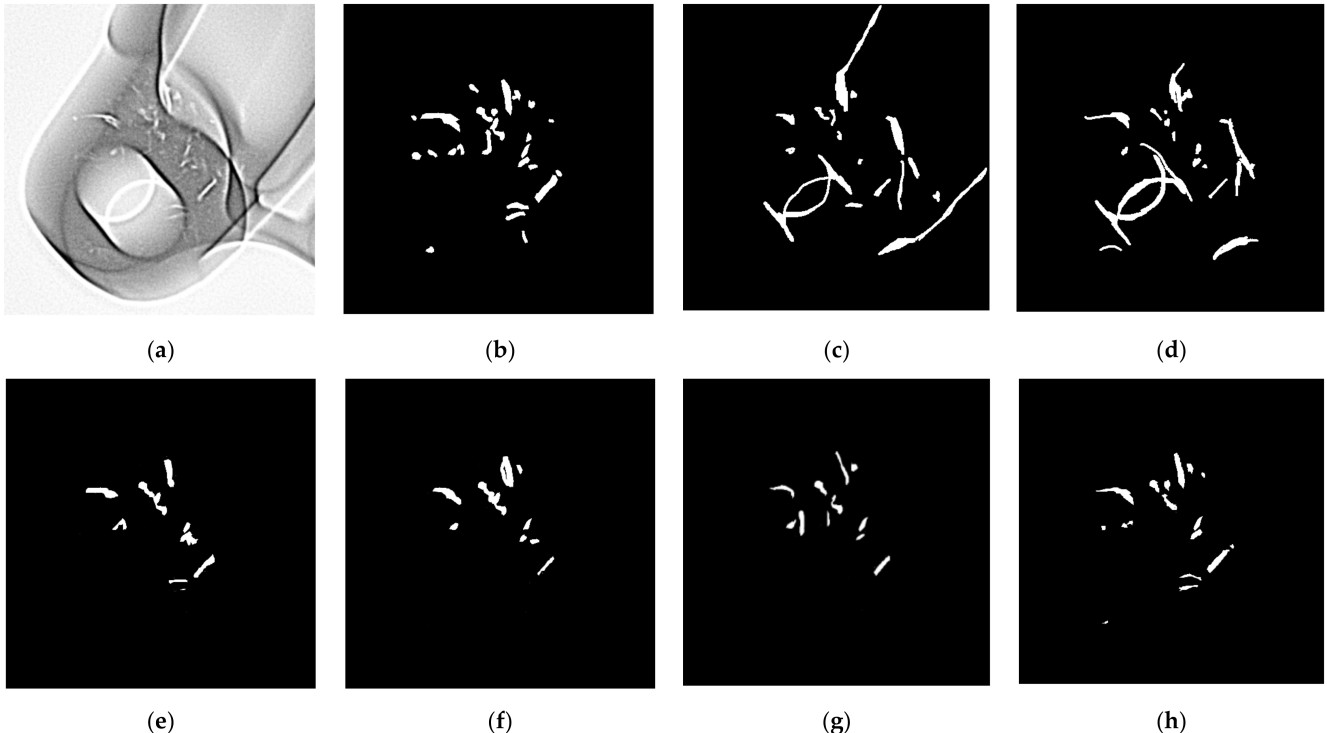

**Figure 10.** A typical example of defect segmentation: (**a**) the original image, (**b**) ground-truth image, (**c**) result of median and morphological filters, Reference [3], (**d**) result of a traditional defect detection algorithm, Reference [4], (**e**) result of Mask R-CNN, (**f**) result of UNet, (**g**) result of standard UNet++, (**h**) result of proposed method.

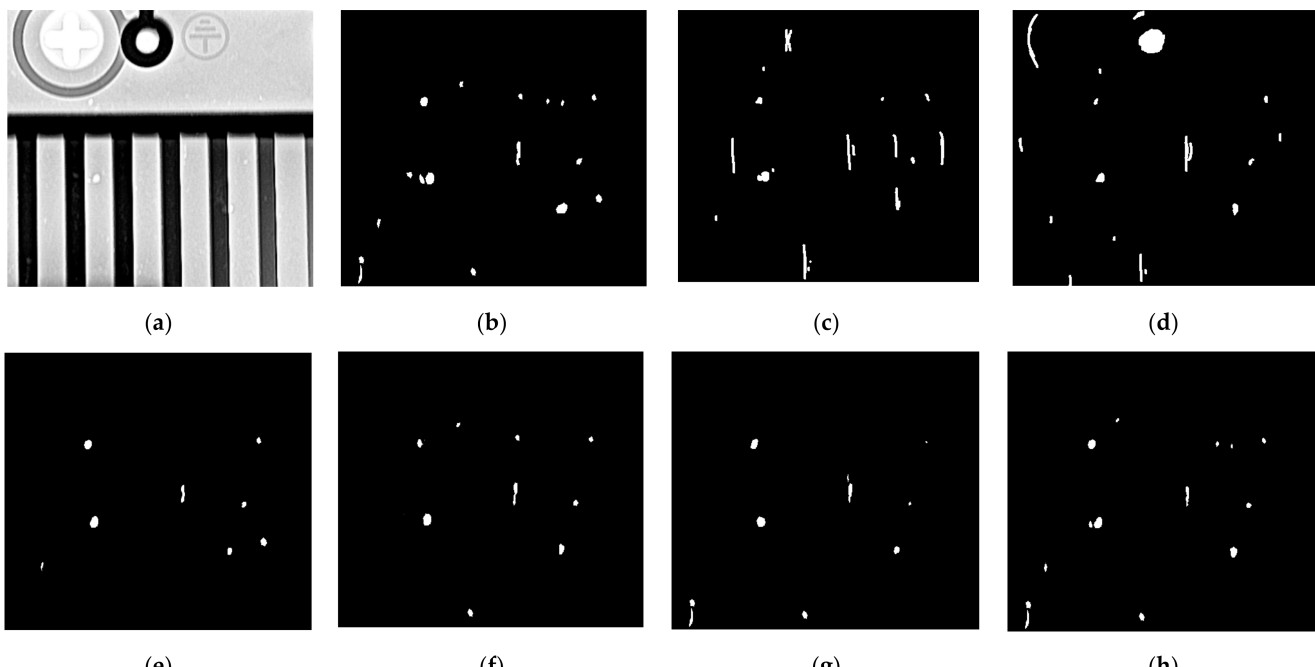

**Figure 11.** Another example of defect segmentation: (**a**) the original image, (**b**) ground-truth image, (**c**) result of median and morphological filters, Reference [3], (**d**) result of a traditional defect detection algorithm, Reference [4], (**e**) result of Mask R-CNN, (**f**) result of UNet, (**g**) result of standard UNet++, (**h**) result of proposed method.

**Figure 12.** Segmentation examples of proposed method. (**a**–**d**) are the original images; (**e**–**h**) are the ground-truth images; (**i**–**l**) are the corresponding segmentation results.

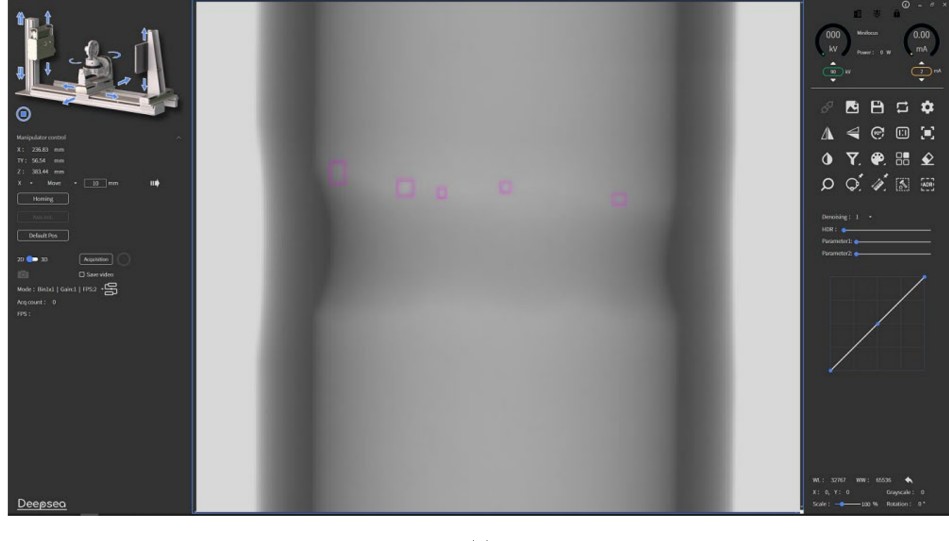

(**a**)

**Figure 13.** *Cont.*

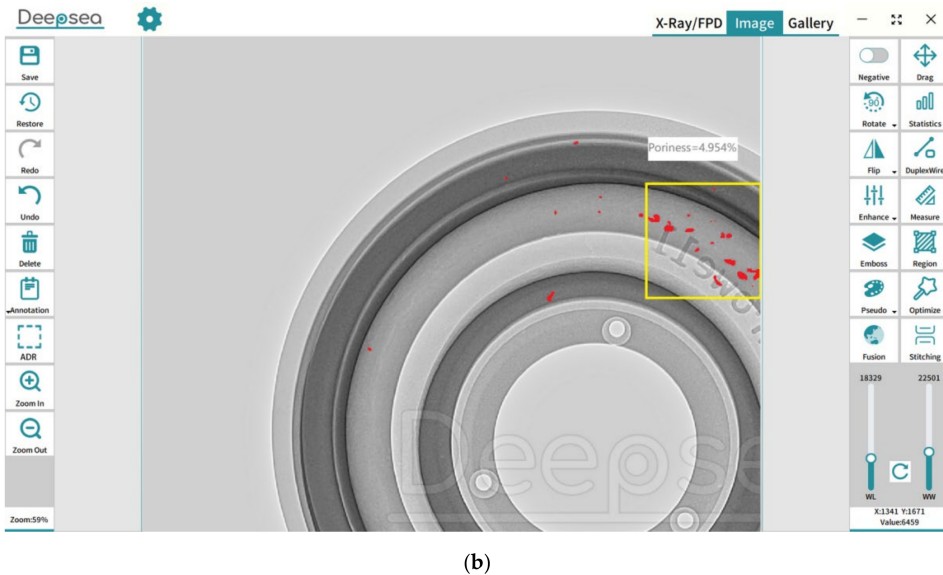

(**b**)

**Figure 13.** Applications of proposed method: (**a**) automatic welding defect detection, (**b**) automatic casting defect detection.

## 4. Discussion

In this section, we will discuss the results obtained by the proposed defect detection system as well as other methods including some deep learning models and conventional defect detection methods. The segmentation results accordingly are shown in Figures 10 and 11. According to our experimental results, deep learning methods provide better segmentation performance in comparison to conventional image processing methods. In general, conventional methods generate high false positive rate due to the characteristics of industrial X-ray images.

Figure 12 shows that our model performs well on complex images and is able to detect different kind of defects. The performance of the proposed model and some other models are evaluated in Table 5. Mask R-CNN and U-Net showed similar accuracy in the experiments. In addition, the proposed modified UNet++ model outperformed the standard UNet++. Due to the improvement of image quality through proposed pre-processing, our defect segmentation method achieves 89.24% and 94.31% in terms of IoU and Dice, respectively.

Although the overall performance of the proposed method has been demonstrated, there are still some missed detections in the experiments. This is mainly because some thin defects are slightly stronger than the background, increasing the difficulty of detection. Moreover, some incorrect detections were also observed due to the similarity between the edges or holes of the parts and the defects.

## 5. Conclusions

In this paper, we describe a new method for X-ray image defect segmentation by introducing a variable attention nested UNet++ network. In comparison with the existing techniques, the proposed algorithm has the following features: (i) a pre-processing method based on pyramid model is proposed to further enhance the performance of the faint defect extraction and its clear visibility; (ii) the traditional convolutional blocks are replaced by SK blocks, and adding attention gates between nested convolutional blocks enables increase in the weight of the target region while suppressing background regions that are not relevant to the segmentation task; (iii) it enables model pruning during testing by introducing deep supervision, which can reduce a large number of model parameters and thus speed up the model segmentation. Moreover, the proposed segmentation method proved reliable in localizing defects from the DSXImage database, achieving IoU and Dice values of 89.24% and 94.31%, respectively.

In the future, we plan to explore robust quantization and model compression techniques to further reduce the model complexity and accelerate the inference process. Moreover, the current proposed defect detection system is developed to identify the defects without classification. In the future work, this system will be modified and improved to identify different types of defects such as gas holes, gas porosity, and shrinkage cavity.

We will continue to push forward in the construction of the database, and with the consent of Deepsea, we hope to make the database public in the near future so that more people can participate in the work, continue to improve the defect segmentation performance, and make the algorithms better serve the industry.

**Author Contributions:** Conceptualization, J.L. and J.H.K.; methodology, J.L.; software, J.L.; validation, J.L.; formal analysis, J.L.; investigation, J.L. and J.H.K.; resources, J.L.; data curation, J.L.; writing—original draft preparation, J.L.; writing—review and editing, J.H.K.; visualization, J.L.; supervision, J.H.K.; project administration, J.L.; funding acquisition, J.L. All authors have read and agreed to the published version of the manuscript.

**Funding:** This research was supported by Guangdong Science and Technology Project, grant number 2020A0505100012, 2017B020210005 and BK21FOUR, Creative Human Resource Education and Research Programs for ICT Convergence in the 4th Industrial Revolution.

**Acknowledgments:** Thanks to Deepsea Precision Co. Ltd. and for their X-ray inspection equipment and support for the large number of samples tested over two years. Thanks to all the people who contributed to the database annotation. Special thanks to Stan, a colleague of J.L., who did a lot of work in organizing the data.

**Conflicts of Interest:** The authors declare no conflict of interest.

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
