# Peer review of "A Variable Attention Nested UNet++ Network-Based NDT X-ray Image Defect Segmentation Method"

_coatings, doi:10.3390/coatings12050634_

Round 1
Reviewer 1 Report
I have the following suggestions:
1, More qualitative results should be added with zoomed-in views for the readers to see the defects clearly.
2, The qualitative results of existing representative methods should also be added for comparison.
3, besides the deep learning based methods, other existing defect/object/image segmentation methods should also be compared.
Reviewer 2 Report
The authors need to take in consideration the following suggestions before to accept it:
- The introduction needs to be improved discussing current articles because only a few new articles have been included in it.
- I recommend presenting a table where the qualitative and quantitative features of your proposal and the other proposals reviewed in order to show the main advantages and disadvantages of your proposal.
- The analyzed datasets must be explained in detail.
- A detailed explanation of obtained results is missing.
- The quality of figures needs to be increased, also the size of labels needs to be increased.
- The conclusion needs to be improved, adding quantitative results not only qualitative results. In addition, it is important to mention, what is the next with the investigation?
Reviewer 3 Report
In this article, the authors described a new method for Non-Destructive Testing (NDT) X-ray image defect segmentation by introducing variable attention nested UNet++ network. The proposed segmentation method proved reliable in localizing defects from the DSXImage database, achieving IoU and Dice values of 89.24% and 94.31%, respectively. The theme of the paper is within the scope of the journal. This topic is very interesting and has practical values. The manuscript is properly organized, and the concepts are well introduced. The analysis is impressive and convincing. The figures are clear and helpful for the readers to understand the ideas. It is an interesting and well-written article. As a result, I would recommend the publication of this paper in its present form.
Author Response
Dear reviewer,
We deeply appreciate the time and effort you have spent in reviewing our manuscript (ID: coatings-1689463). Thank you for your insightful feedback and great help.
Reviewer 4 Report
The paper would benefit from a block diagram of the proposed X-ray image segmentation algorithm at the beginning of Chapter 2, indicating the steps proposed by the authors, which are new compared to the existing algorithms, in terms of clarity and comprehensibility for the reader.
Round 2
Reviewer 1 Report
The paper has been improved with the added qualitative results. I still have the following suggestions.
1, For all the qualitative results, the ground-truths must be added.
2, I think the defects could be robustly segmented by SDD threshold selection [1-2] with proper preprocessing and postprocessing steps. The authors should compare or discuss it.
[1] ZZ Wang,Automatic localization and segmentation of the ventricles in magnetic resonance images, IEEE Transactions on Circuits and Systems for Video Technology 31 (2), 621-631
[2] ZZ Wang,A new approach for segmentation and quantification of cells or nanoparticles,IEEE Transactions on Industrial Informatics 12 (3), 962-971
Reviewer 2 Report
The authors have improved the quality of their work according to the reviewer's suggestions. Hence, I can recommend accepting the article.
Author Response
Dear reviewer,
Thank you again for your time and your great suggestions on our manuscript. On behalf of my co-author, we would like to express our appreciation to you.
Thank you and best regards.